# The Influence of Textile Structure Characteristics on the Performance of Artificial Blood Vessels

**DOI:** 10.3390/polym15143003

**Published:** 2023-07-10

**Authors:** Chenxi Liu, Jieyu Dai, Xueqin Wang, Xingyou Hu

**Affiliations:** College of Textiles & Clothing, Qingdao University, Qingdao 266000, China; 15554710715@163.com (C.L.); djy1586803302@163.com (J.D.); 619304316@163.com (X.W.)

**Keywords:** knitting, weaving, electrospinning, braiding, artificial blood vessels, graft

## Abstract

Cardiovascular disease is a major threat to human health worldwide, and vascular transplantation surgery is a treatment method for this disease. Often, autologous blood vessels cannot meet the needs of surgery. However, allogeneic blood vessels have limited availability or may cause rejection reactions. Therefore, the development of biocompatible artificial blood vessels is needed to solve the problem of donor shortage. Tubular fabrics prepared by textile structures have flexible compliance, which cannot be matched by other structural blood vessels. Therefore, biomedical artificial blood vessels have been widely studied in recent decades up to the present. This article focuses on reviewing four textile methods used, at present, in the manufacture of artificial blood vessels: knitting, weaving, braiding, and electrospinning. The article mainly introduces the particular effects of different structural characteristics possessed by various textile methods on the production of artificial blood vessels, such as compliance, mechanical properties, and pore size. It was concluded that woven blood vessels possess superior mechanical properties and dimensional stability, while the knitted fabrication method facilitates excellent compliance, elasticity, and porosity of blood vessels. Additionally, the study prominently showcases the ease of rebound and compression of braided tubes, as well as the significant biological benefits of electrospinning. Moreover, moderate porosity and good mechanical strength can be achieved by changing the original structural parameters; increasing the floating warp, enlarging the braiding angle, and reducing the fiber fineness and diameter can achieve greater compliance. Furthermore, physical, chemical, or biological methods can be used to further improve the biocompatibility, antibacterial, anti-inflammatory, and endothelialization of blood vessels, thereby improving their functionality. The aim is to provide some guidance for the further development of artificial blood vessels.

## 1. Introduction

Cardiovascular disease (CVD), mainly ischemic heart disease (IHD) and stroke, is the leading cause of human health problems worldwide. According to an assessment and statistical analysis using the Global Burden of Disease (GBD) from 1990 to 2019 in 204 countries and regions, it was found that the total number of cases of CVD has almost doubled in quantity from 1990 to 2019, reaching a staggering 523 million cases, with a death toll of 18.6 million people [1]. CVD has become the main cause of disease burden in the world, with annual patient costs exceeding USD 300 billion [2]. To date, arterial disease is often resolved through autologous vascular bypass surgery; however, postoperative risks are significant and may lead to acute thrombosis formation, neointimal hyperplasia, and accelerated atherosclerosis, resulting in graft failure [3]. This can cause further damage to patients, who may experience wound infections and complications, as well as frequent discomfort, leading to prolonged hospitalization and medical expenses [4,5]. In addition, the shortage of allogeneic donors and the difficulty in preserving and manipulating them are also major obstacles to the treatment of CVD [6,7]. Therefore, it is urgent to conduct research on artificial grafts and materials to address these issues.

Since the 20th century, researchers have attempted numerous materials and manufacturing methods to prepare artificial blood vessels. To date, there are many methods for manufacturing, including 3D printing, gas foaming, hydrogel tube formation, textile weaving, among others. Three-dimensional printing technology has been popular for decades and is used for rapid artificial blood vessel production. By using computer software to print thin-layer materials layer by layer, a 2D image can be transformed into an actual 3D model [8]. The advantages of this technology include rapid customization and clear personalization; however, it cannot yet be directly used for transplantation in patients [9]. The research at present mainly focuses on the in vitro production of vascular models, preparation of epidermal cell-related structures, and microvascular networks. Gas foaming methods (including batch foaming, micro-injection molding, and extrusion foaming) are used to produce artificial grafts mainly by utilizing gases (CO_2_ or N_2_) to nucleate and expand porous structures [10,11]. For example, a PCL/PLA foam scaffold can be used as a vascular graft. However, this method produces objects with poor closed-cell and interconnectivity, and a low bending modulus, which limits its application. Hydrogels have good biocompatibility but low mechanical strength and are difficult to process alone; therefore, they need to be combined with other methods, such as electrospinning [12].

For artificial blood vessels, simulating the biomechanical properties of natural grafts and human tissue is an important requirement. Textile artificial blood vessels use primitive fibers, single yarns, or strands that are interwoven in a regular interlace or irregular arrangement to form medical implant vessels with high mechanical strength. This creates a multifunctional product with excellent mechanical and biological properties that largely mimic the structure and functionality of biological tissue [5]. Blood vessels created using this method can be similar to human tissue, making it easier for cells to adhere and grow, which is difficult to achieve through other fabrication methods. In terms of the design, textile grafts have more flexible properties than other materials. Ideal effects can be achieved by changing the fiber fineness, yarn structure, fabric organization, and other means. In terms of flexibility, at the end of the 20th century, researchers used stainless steel (SS) or a cobalt–chromium (CoCr) alloy to produce bare metal stents, and found that, although they have a strong support force, the grafts are prone to lesions or inflammation, and have poor adaptability [13,14]; textile structures have remarkable effects in the field of soft tissue repair, and are more flexible than metal or plastic. Structurally, textiles can be precisely designed in 2D and 3D spaces, with unique interweaving and highly porous structures, making textile grafts have good permeability, providing a basis for cell migration and tissue regeneration [15]. Figure 1 shows the main manufacturing processes used at present for artificial grafts: knitting, weaving, braiding, and electrospinning. It is worth noting that many blood vessels are not manufactured using only one of these methods, but sometimes combine multiple methods to adapt to the human body and achieve better mechanical properties or optimal structures.

To date, there are many materials available for preparing artificial blood vessels, which can be divided into natural and synthetic polymers (Table 1). The commonly used natural polymers are polysaccharides, collagen, fibrinogen, chitosan, elastin, and cellulose. Researchers have found that artificial grafts made from natural polymers have better biocompatibility, which can promote the proliferation of endothelial cells, help to maintain the graft’s patency, and reduce the thrombogenicity [16,17,18]. Natural polymers degrade rapidly and uncontrollably, which can lead to the loss of mechanical properties after implantation and affect the long-term use of artificial blood vessels. Synthetic polymers, such as polycaprolactone (PCL), polylactic acid (PLA), thermoplastic polyurethane (TPU), polyglycerol sebacate (PGS), and poly(lactic-co-glycolic acid) (PLGA), exhibit good mechanical properties and biostability, with slow and controllable degradation rates [19]. However, the biocompatibility of synthetic polymers is poorer than that of natural polymers, with slower endothelialization and poorer long-term patency of after implantation [20,21,22].

Although textile structures have been widely used in the development of large-diameter artificial grafts, they still have problems, such as a slow endothelialization rate, low long-term patency rate, and thrombosis and restenosis, in the use of small-diameter artificial blood vessels. Therefore, modifications are needed to improve their biocompatibility. This can be conducted from the aspects of improving the endothelialization degree, biocompatibility, antibacterial and anti-infection properties, thrombus obstruction, etc. These goals can be achieved through physical and chemical means, such as optimizing raw materials, coatings, and employing innovative techniques [23].

This article reviews four preparation methods for textile-based artificial blood vessels. The structural differences in these grafts obtained through different textile techniques are described, with a detailed analysis of the compliance, mechanical strength, porosity, and fabric structure. Furthermore, optimization methods related to the required performance of grafts, such as antibacterial properties, infection resistance, biocompatibility, and degree of endothelialization, are summarized, in order to develop textile-based artificial grafts with a better performance.

## 2. Knitting

The application of knitted-structure artificial blood vessels has a certain history; as early as 1955, it began to enter clinical applications [24]. Knitting is formed by interlocking loops of yarn, and the knitted structure is softer and more compliant than woven structures. Different materials can be used to make yarns, such as natural and synthetic materials. Figure 2 shows the two types of knitted-structure artificial grafts: warp and weft knitting vessels. Warp knitting is a type of knitted structure where yarns are fed in the warp direction and sequentially laid on top of each crochet needle on a warp knitting machine, before being formed into a fabric through coordinated machine operations. In contrast, weft knitting involves multiple yarns being simultaneously fed in the weft direction and laid on top of the crochet needle to form a fabric on a weft knitting machine.

Knitted artificial grafts have gained many advantages due to their unique interlacing method and structure. With a looped structure, knitted fabrics exhibit excellent extensibility, elasticity, and good conformity and adaptability. Due to the pore structure of the knitted fabric, it has good permeability, breathability, moisture conductivity, and high porosity. For implanted artificial grafts, this is conducive to material exchange inside and outside the vessels and promotes cell migration. However, textile fabrics with a large-pore-size structure allows for easier blood effusion compared to materials obtained through other preparation methods [25]. Moreover, the in-plane strength of the knitted structure is low, and it is still difficult to create a structure with adjustable properties in different directions by knitting [12].

Knitted blood vessels have been widely used for tissue engineering or repairing damaged tissues, and many teams have conducted related research on knitted grafts, evaluating their advantages of long-term use and durability [26,27]. A team once studied a biodegradable PGA tubular knitted stent and seeded an endothelial cell layer on it. The final results showed the excellent durability of the knitted blood vessel [28]. Meanwhile, another experiment designed an artificial blood vessel with a knitted PET and collagen composite structure inside and woven PLA outside, and performed a dynamic evaluation in vivo for 12 months to prove its durability [29]. Although some blood vessel products based on knitted structures are more or less combined with biopolymers or seeded with an endothelial cell layer, the knitted structure can facilitate the biological performance of the abovementioned substances, playing a considerable role in the long-term stability of grafts. The reason is that knitting, as a textile structure, has mechanical compatibility with the human body (which reduces abnormal inflammatory reactions of grafts) and the unique pores provide significant convenience for the biological communication between cells (which will be described in more detail below), coupled with the appropriate mechanical strength of knitted vessels, which helps them perform well from multiple aspects [30,31,32]. Although the defects of knitted blood vessels still need to be improved, they are still an unavoidable focus of consideration in the research and development of tissue-engineered blood vessels.

### 2.1. The Impact of Knitted Structure on the Performance of Artificial Blood Vessels

#### 2.1.1. Diversity in Mechanical Properties of Knitted Artificial Blood Vessels

The mechanical performance of knitted blood vessels varies depending on their structure. In this paper, we explored the mechanical differences resulting from different knitting techniques (warp and weft) and how they affected the deformation of the grafts.

In knitted artificial grafts, the mechanical performance of the warp knitting structure is more stable than that of the weft knitting structure. In their study on the manufacturing and performance of PVA knitted grafts, Jia-Horng Lin et al. evaluated the circumferential mechanical properties of tubular weft knitted and warp knitted fabrics, demonstrating that both structures exhibited good compressive properties, with the warp knitted structure exhibiting a superior circumferential mechanical performance. They also optimized and modified the structure and function of the tubular fabrics [33,34,35]. Therefore, in the preparation, it was possible to select the appropriate warp and weft knitting structures according to the specific mechanical performance requirements.

#### 2.1.2. Knitted Porous Structures Facilitate Intercellular Substance Exchange

The knitted structure has a higher porosity. Tests have shown that the porosity can reach 63–84%, which is several tens of percentage points higher than that of woven vessels [36]. Of course, as the number of fabric layers increases, the porosity decreases. Many scholars believe that artificial grafts require sufficient porosity to promote nutrient exchange and vascular endothelialization to help maintain long-term patency and the better biological performance of grafts [36,37]. Knitted materials have advantages in terms of porosity; however, a higher porosity can lead to significant blood effusion during the transplantation. Therefore, pre-coagulation before surgery or applying a sealing material coating during the manufacturing process is generally required. However, pre-coagulation may cause residual blood clots, and adding coatings with good biocompatibility has become a recent focus of the research. Some scholars have optimized knitted grafts with materials, such as gelatin, SF, and fibrin glue, to better control blood effusion [38,39,40]. Scholars utilized a double-raschel knitting machine to produce 1.5 mm knitted tubes, onto which four different concentrations of SF coatings were applied. It was found that the material coated with the 7.5% concentration was more effective in inhibiting endothelial proliferation and vascular restenosis [41]. Therefore, applying a silk coating can effectively prevent blood effusion. The high porosity of knitted structures in blood vessels provides assistance to cellular growth and substance exchange within tissues. However, the existing research has shown that the porosity of knitted materials has not been effectively utilized due to concerns about blood effusion. The challenge remains of how to maximize the rational utilization of the high porosity of knitted structures.

#### 2.1.3. Knitted Artificial Blood Vessels Exhibit Excellent Compliance

Knitted artificial blood vessels exhibit superior overall compliance compared to other types of artificial blood vessels. Although there is little difference in radial compliance compared to braided grafts, it is superior to woven materials. From the perspective of the interlacing method of knitted blood vessels, although the number of warp knitted grafts is relatively high, the compliance of weft knitted grafts is superior to that of warp knitted grafts. This is mainly due to the different loops forming way. Weft knitted artificial grafts have good stretchability, both circumferentially and axially, resulting in better compliance.

Regarding the knitting structure, changes in the fabric structure can also have a certain impact on the compliance of knitted grafts. Taking the warp knitting artificial blood vessel as an example, to date, the tricot stitch, reverse locknit, and atlas stitch structures are commonly used; however, their compliance varies greatly. Researchers preparing double-raschel knitted silk grafts used three types of warp knitting structures as controls. The torsion resistance test showed that the torsion radius decreased in the order of the double-tricot stitch, the reverse locknit stitch, and the combination of tricot and atlas stitches. This indicated that the double-tricot stitch structure was more elastic, flexible, and inhibited intimal hyperplasia, while the other two structures exhibited relatively weak adaptability [42]. How to better utilize warp knitting structures to improve vascular compliance is still a subject that requires further exploration.

### 2.2. Modification of Knitted-Structure Artificial Blood Vessels

#### 2.2.1. Modification of Biocompatibility for Knitted-Structure Blood Vessels

Early studies found that, compared to some smooth surfaces (such as polystyrene and PET film), the adhesion and growth rate of endothelial cells on knitted artificial blood vessels were poorer [43]. Therefore, in order to enhance the adhesion ability and endothelialization rate of endothelial cells on the surface of knitted artificial grafts, various modification methods need to be conducted to improve their biocompatibility. Commonly used methods to enhance the biocompatibility of knitted artificial grafts include in vitro pre-endothelialization culture, using biocompatible yarns to prepare implanting endothelialization-promoting factors onto the surface of the grafts.

In the biomedical field, obtaining human endothelial cells, isolating and culturing them, and then seeding them onto artificial grafts can facilitate endothelial cell proliferation and coverage. A research group isolated, cultured, and characterized EPCs, and seeded them into collagen-infused knitted PET artificial grafts. Through the evaluations of cell toxicity, proliferation, attachment, and retention on exposure to flow, it was found that MMP1 and mRNA increased within 4 h, while vWF, VE-cadherin, and KDR did not show significant changes at 4 and 8 h, indicating that PDEC derived from umbilical cord blood could form an effective endothelium and enhance anti-shear stress ability [44]. The research on the potential applications of EPCs in artificial grafts started as early as 2001 [45]. This modification method will have a considerable impact on the fields of cell therapy and regenerative medicine and can be considered as a more complex and biologically based modification method.

Another group of researchers developed a knitted artificial blood vessel scaffold by blending biocompatible PLA yarns with collagen filaments. Measurements were taken for indicators, such as endothelial cell adhesion, suture retention strength, and suture tensile strength. It was found that collagen was a favorable promoter for endothelial cell aggregation and proliferation, which can increase initial cell adhesion by 10 times. The final cell population of this two-component mixed knitted vessel was 3.2 times higher than that of the PLA single-component vessel, and it also exhibited a good mechanical performance [46]. There are also related experiments that used biodegradable PVA yarn to create vascular stents, which exhibited good biocompatibility [35]. Using biological yarns as raw materials for knitted graft preparation enhances the biological properties of blood vessels from the source and better utilizes the characteristics of knitted structures. However, the use of biological yarns, such as collagen, requires a consideration of strength and stability to prevent surgical risks.

If we focus on the addition of endothelialization-promoting factors, attention can be paid to small-diameter knitted blood vessel coatings made by De Visscher et al. They use fibronectin and stem cell homing factor SDF-1α, which were found to improve the endogenous endothelialization of grafts. Through in vivo experiments in sheep, it was found that the coating stimulated endothelial cell proliferation and adhesion, and effectively reduced intimal hyperplasia and thrombosis [47]. Compared to the cell seeding modification method mentioned earlier, this type of method avoids the complex procedure of cell collection, reduces infection risks, and avoids the waiting period for cell culture.

In addition, Obermeier et al. made a new attempt in vitro: modifying flax-knitted blood vessel structures with an albumin coating, hoping to use the advantages of flax’s hydrophilicity and strength to prepare knitted artificial grafts with excellent mechanical properties and biocompatibility. The experiment showed that the albumin coating helped reduce the effect of flax toxins and promoted endothelial cell adhesion [48].

Although it is difficult to achieve a better biological compatibility of knitted artificial blood vessels solely through material selection or vascular structure design, biological properties can be enhanced by using degradable and collagen yarns, or by post-treatments, such as cell seeding, the addition of biological factors, and various types of coatings. To date, there are multiple methods available for reference, and the selection of specific methods and the combination of multiple methods depend on the actual application scenario.

#### 2.2.2. Modification of Knitted Vascular Structures for Antibacterial and Antimicrobial Properties

Infection of artificial blood vessels mainly refers to the adhesion and proliferation of bacteria during transplantation surgery, eventually forming an irreversible attachment on the surface of the artificial blood vessel, leading to the formation of dense microbial communities. It can easily lead to transplant failure and is one of the reasons for hindering the long-term patency of grafts [49]. Therefore, the antibacterial properties of artificial blood vessel materials should be considered in their design and preparation. Moreover, the main bacteria observed in infections at present are Methicillin-resistant Staphylococcus aureus and Staphylococcus epidermidis [50,51]. In order to enhance the antibacterial properties of knitted artificial grafts, drug-loaded polymer coatings can be used for the enhancement of antibacterial properties.

For the modification effect of drug release, a team conducted an assessment on a novel sustained-release antibiotic coating made of modified neomycin fatty acid salts. The method utilized gentamicin as a representative of preferred antibiotics, which can effectively eliminate Staphylococcus and aerobic Gram-negative bacilli. It was found that the coating could continuously release drugs for the first eight hours, and gentamicin palmitate exhibited the best anti-infection efficiency, biocompatibility, and blood compatibility among the three substances in the control experiment, reducing the probability of artificial blood vessel infection. To date, many modification methods use polymer films attached to the surface of blood vessels to treat inflammation and resist bacteria; however, some films have an adverse effect on the flexibility and stretchability of grafts. Thus, there are new studies devoted to improving the properties of coatings [52,53]. Al Meslmani et al. prepared a multifunctional network-structured film that maintains the basic elastic and folding ability of knitted grafts. They used an antibacterial sulfadimethoxine polyhexylene adipate-b-methoxy polyethylene oxide (SD-PHA-b-MPEO) di-block copolymer as a coating. The mechanism of the coating was to utilize the negatively charged, antibacterial sulfadimethoxine group and the bacterial cell-repelling characteristics of the hydrophobic PHA and MPEO blocks to inhibit bacterial adhesion. After the antibacterial treatment, the test results showed that the adhesion of Gram-positive Staphylococcus epidermidis in the knitted material was reduced by 2.3 times, the anti-adhesion ability of Gram-positive Staphylococcus aureus was 4-times greater than before, and the anti-adhesion effect towards Gram-negative Escherichia coli was 2.7-times greater. The antibacterial effect was significant. The continuous development of the abovementioned modification method significantly improved the antibacterial performance of the knitted-structure artificial grafts.

## 3. Weaving

As early as 1950, woven structures were employed in the manufacture of artificial blood vessels. The fabrics shown in Figure 3 are woven fabrics made by interlacing warp and weft yarns at a 90° angle according to a specific weaving structure, which can be mainly divided into three types of structures: plain, twill, and satin weaves.

The plain weave is the most commonly used tissue in the fabrication of woven artificial blood vessels. It is the simplest structure, formed by interweaving the warp and weft yarns over and under each other alternately. The plain weave has a high number of yarn intersections, resulting in a smooth surface and superior mechanical properties, making the fabric more durable. In contrast, the twill weave differs from the plain weave in the way that the warp and weft yarns are interwoven, resulting in diagonal ridges on the fabric surface (which can be represented as a fraction, such as 2/3↗ to indicate a right-leaning twill with two warp and three weft intersections per complete repeat). Although the stability and firmness of this structure are not as good as a plain weave, the resulting fabric is softer. Satin weave has a longer distance between adjacent warp or weft yarns on the fabric, and longer floating yarns. Its weave points are arranged according to a certain pattern, with at least five warp and weft yarns in a complete structure. Due to its longer floating yarns, it has the lowest strength among the three types of fabric weaves, but is the softest.

Due to the intersecting structural characteristics of the warp and weft, woven fabrics have a low porosity with a regular pore distribution and stable pore size and shape [54]. In particular, compared to knitted fabrics, the low porosity of woven fabrics facilitates reduced blood effusion during implantation. Additionally, woven structures exhibit good surface uniformity and area density. In addition, woven fabrics exhibit excellent mechanical strength in textile-based artificial grafts, with little deformation and good dimensional stability. However, they may have negative effects that lead to decreased vascular elasticity and weakened compliance, which is a disadvantage when compared to knitted fabrics [15].

### 3.1. The Impact of Woven-Structure Characteristics on Artificial Blood Vessels

#### 3.1.1. Optimized Design Required for Low Compliance

Due to the different interlacing methods of yarns, woven fabrics show poorer extensibility and elasticity than knitted fabrics, which, to a certain extent, results in low compliance. It is concerning that if the compliance of the fabric is not consistent with that of human blood vessels, implantation may lead to the formation of eddies in the graft, which could result in thrombosis and subsequent restenosis, ultimately affecting the long-term patency of the blood vessel.

The compliance of textile structures still has considerable room for improvement in vascular applications. In woven vascular grafts, compliance shows an inverse correlation with the size, density, and Young’s modulus of warp and weft yarns: the thicker the yarns, the denser the fabric, and the greater the tensile and compressive strength, the lower the compliance [55]. If optimization is achieved from the fabric itself, using fabrics with a longer float length, such as twill or satin weaves, can result in better compliance [55]. On the other hand, it is worth noting the compliance matching between woven grafts and human vascular structures and tissues, and the development of multilayered artificial grafts can enhance this capability to some extent. A team developed dual-layered woven grafts using a PTT filament as the inner layer and a PET filament as the outer layer. The two layers have different tensile moduli and are connected by suture filaments. This structure can enhance the radial conformability of the inner layer and the durability of the outer layer. As a result, the compliance of the woven grafts is enhanced by 2 and 1.4 times under high and low systolic pressures, respectively, compared to single-layered PET woven grafts. This has a positive effect on improving the radial compliance of the woven grafts [56].

In addition to the inherent low compliance of woven structures, inflammation and tissue proliferation caused by immune rejection after implantation can also make the blood vessel wall stiff and affect its compliance. There are reports of making modifications in the material aspect. Researchers used supercritical N-2 (SCN2) jet technology to increase the roughness of yarns with various diameters (100 and 400 μm) involved in woven vascular grafts, and the results show that this technology has a certain inhibitory effect on the proliferation of human fibroblasts, which is an innovative approach [57].

Although woven artificial grafts have inherent lower compliance compared to knitted ones, their adaptability within the human body can be improved through design and structure, layers, or materials.

#### 3.1.2. The Influence of Fabric Structure on the Performance of Vascular Textiles

Different fabric structure designs of woven blood vessels can result in varying textile structures and consequently different performance characteristics [58].

The plain weave structure is more favorable for the endothelialization of implanted artificial grafts because the extracellular matrix (ECM) relies more on a tight and flat surface [59]. Furthermore, fabric density is the most important textile parameter affecting the performance of artificial grafts, while yarn count affects the fabric density. According to the study, implants made of deformable polyester yarns with a plain weave structure were better. The warp and weft linear densities of 110 and 167 dtex and a warp density lower than 26 ends/cm exhibited excellent stiffness, resistance, and appropriate porosity. Furthermore, since the circumferential strength mainly relies on the support of the weft yarn, the weft yarn density is generally higher than the warp yarn density. To obtain a fabric with a low risk of rupture, the following parameters can be used: a warp linear density less than 130 dtex, weft linear density higher than 167 dtex, and warp density of ≤25 ends/cm [58].

Under certain conditions, twill weave artificial blood vessels exhibit good mechanical properties. In the relevant experiments, a narrow-ribbon shuttle loom was used to weave small-diameter woven artificial grafts using 179D/108F polyester and 204D silk protein yarns. Three weaving methods were used: 1/1 plain weave, 2/2 twill weave, and 1/3 twill weave. The final performance test indicated that the 2/2 twill structure exhibited the best performance. In actual blood vessel design, the twill structure can be considered as a reference, while taking into account the influence of yarn size and material to achieve better results.

Satin weave fabrics have a high porosity and exhibit high hydrophilicity. This low-saturation woven fabric restricts the foreign body reaction of cell overproliferation to some extent; plain weaves display the opposite effect [24]. Therefore, in the preparation of artificial grafts, the use of satin weave structures can be considered to limit excessive cell proliferation [60]. However, the satin weave is not commonly used, and further research in this area is needed.

#### 3.1.3. Porosity and Vascular Effusion

The porosity of woven artificial blood vessels is relatively lower compared to that of knitted fabrics. Although the smaller pores of woven vessels have a weaker influence on cell adhesion and proliferation, they can reduce blood effusion and minimize tangling, making them more suitable for applications in large-diameter and high-blood-flow vessels.

Textile structured artificial grafts have a regular and controllable porous structure that cannot be achieved by other non-textile structured artificial grafts. However, at present, the clinical application of porous textile-based artificial blood vessels is not sufficient, as the concern for graft leakage leads to a near-zero water permeability of the artificial blood vessels [61]. A study investigated four types of woven blood vessels to explore the permeability of textile grafts and simulate plasma permeability. It found that it is possible to utilize the good permeability advantages of textile structures to a certain extent, rather than just pursuing lower permeability results [61]. The optimal balance between the porosity and permeability of woven artificial grafts and the prevention of graft bleeding remains unclear. A further exploration of and development in this area are necessary.

### 3.2. Modification of Woven Artificial Blood Vessels

In recent years, there have been many studies on the modification of woven artificial blood vessels using SF, and some researchers combined SF with polyester to produce two-component artificial grafts [62,63]. These attempts effectively improved the mechanical properties and biocompatibility of woven artificial grafts [64]. In terms of innovative research, Agathe Grémare and her team started with biological yarn materials, developed yarns composed of human amniotic membrane (HAM), and combined them with weaving technology to prepare artificial grafts, aiming to improve the biocompatibility of artificial blood vessels. This paper described the production of a woven HAM artificial blood vessel with an inner diameter of 4.4 ± 0.2 mm, using a custom weaving machine that twisted the fibers at rates of 5, 7.5, and 10 turns per centimeter. HAM possesses desirable biological properties, such as low immunogenicity, anti-inflammatory effects, antimicrobial activity, and hemocompatibility, which contribute to excellent biocompatibility when used as a graft material. HAM blood vessels also exhibited impressive mechanical strength, with an average burst pressure of 5628 ± 667 mmHg and suture retention strength of 5.30 ± 0.95 N [65]. Perhaps the use of biomaterials can inspire the modification of knitted, woven, and electrospun blood vessels.

## 4. Electrospinning

Electrospinning was first utilized in 1887 and later became a method for producing nonwoven fabrics, applicable to industrial, biological, and tissue engineering technologies [66,67,68,69,70]. In recent years, it has been applied in the preparation of artificial blood vessels and has shown good results in the preparation of small-diameter blood vessels. Unlike the previous manufacturing methods, electrospinning is difficult to be used for the independent preparation of artificial blood vessels. It is generally achieved through the mixing of multiple polymers or the compounding of multiple layers of polymers, as well as in combination with other manufacturing methods.

The polymers used for preparation are mainly synthetic and natural polymers. To date, PLA and PGA are widely used in the field of biomedical applications, and PLA has the characteristic of slow degradation [71]. PLGA is a copolymer of the two aforementioned polymers, and its degradation rate can be better regulated by controlling the ratio of the two substances. Additionally, it exhibits excellent biocompatibility [72,73]. PCL exhibits excellent biological performance, degradability, and non-toxicity results. The abovementioned materials are all synthetic polymers, which may have better strength compared to natural polymers. While these advantages are favored, the researchers also noticed the issues of the biostability in PGA, PLGA, PLCL, and PU, as well as the mismatch in the hydrophobicity and mechanical properties caused by PLA and PCL. Natural polymers, such as collagen, silk fibroin, and polysaccharides, are highly regarded due to their natural and biological properties, despite their mechanical shortcomings. In addition, they are beneficial for cell adhesion and migration, and present advantages, such as good biodegradability, certain strength, biomimetic, and non-toxicity [74,75,76,77,78]. By imparting synthetic polymers with certain bioactive factors, or by mixing synthetic polymers with natural polymers in the process of preparing mixed spinning solutions, the advantages of both are maximized.

Figure 4 illustrates the electrospinning process for vascular fabrication, which involves the polymer fluid being split into tiny polymer jets by electrostatic atomization, and then solidifying into fibers after traveling a certain distance. This process involves the use of electric fields to stretch and elongate the polymer solution. Additionally, in the production of artificial grafts, a rotating mandrel is commonly utilized as a collector. During the final stage of production, due to rapid solvent evaporation, nanoscale diameter fibers are collected onto the mandrel and serve as raw materials. When certain parameters of electrospinning are adjusted, such as the voltage, distance between needle and collector, needle diameter, and flow rate, fibers with different diameters ranging from several nanometers to several micrometers can be produced [79]. In general, fibers with larger diameters may be produced when higher flow rates or more liquid are produced, while smaller fiber diameters are produced at higher voltages. Moreover, increasing the distance between the needle and collector can also result in thinner fibers [80,81]. The variation in the fiber diameter can also adjust the porosity of artificial blood vessel surfaces.

Electrospinning technology has gained widespread use in the fabrication of artificial blood vessels due to its ability to create fibers with unique diameters and porosity outcomes, as well as its excellent biocompatibility and multi-functional capabilities. Polymer solutions are electrospun into nanofibers, which are then collected and formed into a mesh structure that mimics the natural extracellular matrix (ECM). This approach is crucial for enhancing the biocompatibility of artificial grafts and is, at present, only achievable through decellularized scaffolds in addition to electrospinning. Furthermore, the distinctive electrospinning technique allows for the production of multifaceted nanofibers by utilizing a combination of natural polymers, synthetic polymers, and bioactive agents. This method not only generates a biological network-like matrix, but also further increases the biological activity of the resulting material [82,83]. In addition, multiple teams utilized electrospinning to manufacture multilayer structures to further simulate the biological characteristics of artificial grafts. The examples include multilayer artificial grafts that resemble natural elastic tissue (with circularly aligned wavy fiber structures) and three-layer composite-structure artificial grafts composed of fibers and yarns (which improve biocompatibility and mechanical properties) [84,85]. The aforementioned unique advantages greatly augment and stabilize the development and application of electrospinning technology in the cardiovascular domain. Notwithstanding its numerous merits, the pore-size distribution in non-woven electrospun grafts cannot be meticulously regulated to achieve a high degree of regularity and precision, and their mechanical properties are somewhat inferior to those of woven artificial blood vessels [86].

### 4.1. The Impact of Characteristics of Electrospun Structures on Artificial Blood Vessels

Optimizing the process parameters during electrospinning can lead to varying fiber diameters, which, in turn, affect the porosity, fiber orientation, and surface morphology of artificial blood vessels. These variations can significantly impact the vascular structure and ultimately affect the performance of artificial grafts. The subsequent sections provide a detailed analysis of these effects across three key aspects.

#### 4.1.1. The Impact of Fiber Diameter on Endothelial Cell Activity

The impact of electrospun artificial blood vessels on endothelial cell activities is highly dependent on the diameter of the fibers used. Studies have shown that larger fiber diameters promote macrophage polarization into the M2 phenotype [87]. However, increasing the fiber diameter in the matrix was found to decrease endothelial cell adhesion efficiency, as well as their migration ability on the matrix [88]. While the relationship between endothelial cell activities and fiber diameter is apparent, it is not strongly regular. The effect of different fiber diameters on endothelial cells in electrospun (ε-caprolactone) vascular stents was explored, revealing that the surface arrangement of endothelial colony-forming cells (ECFCs) and mature human umbilical vein endothelial cells (HUVECs) is different on matrices composed of large-diameter fibers (5, 8, and 11 μm) compared to those composed of small-diameter fibers (2 μm) [89]. Similarly, Young-Gwang Ko et al. investigated the effect of different fiber diameters on endothelial cells in electrospun biodegradable PLGA artificial grafts. After seven days, a monolayer of cells grew on the surface of 200 nm diameter fibers; however, it did not penetrate the pores. From days 7–14, cells grown on 600 nm and 1.5 μm diameter fibers were able to migrate into the pores compared to those grown on 200 nm diameter fibers. The cell proliferation rate was lowest on 5.0 μm fiber diameter. These results suggest that fibers with a diameter of 600 nm are suitable for use in artificial grafts, as they can improve the biocompatibility of the vessels [90]. Therefore, changing the fiber diameter is a key aspect that affects the biological properties of electrospun blood vessels, and is an important manifestation of the structural impact on performance.

#### 4.1.2. The Effect of Pore Size and Porosity on Vascular Effusion and Endothelialization

The pore size and porosity are influenced by the fiber diameter, which, in turn, affects vascular effusion and endothelialization [91].

PCL grafts with larger pores and thicker fibers were found to enhance cell migration and ECM secretion, as well as induce M2 macrophage migration into graft walls, thereby promoting cell migration and vascularization. Additionally, PCL was shown to contribute to stable mechanical strength due to its slow degradation rate. The data regarding the pore size show that, when the material increases from a pore size of 1 μm and a fiber diameter of 0.4 μm to a pore size of 15 μm and a fiber diameter of 3 μm, the porosity increases from 69% to 83%, indicating a certain correlation between pore size and porosity [92]. However, continuously increasing porosity is not feasible. The pore size needs to be controlled within a suitable range, as excessively large pores create large distances between the fibers, making it difficult for cells to cover the gaps, which is disadvantageous for preventing blood vessel effusion. To address this issue, a bilayer electrospun artificial blood vessel with different pore sizes on the inner and outer sides was developed. The study showed that placing a small-pore-size layer on the inner side and a large-pore-size layer on the outer side of the artificial blood vessel can simultaneously reduce blood effusion and promote endothelialization. Conversely, the opposite effect occurs when the layers are reversed. Another approach to improving pore size and porosity involves producing a polymer material under conditions of glass transition temperature or higher, which causes the surface of the material to relax and become dispersedly porous [93].

For blood vessels in general, the voids contained within the pore walls of the vessel must be large enough to allow for cell migration; therefore, the pore size should be larger than 10 μm [94,95,96]. The pore size of electrospun vascular grafts is influenced by fiber diameter and jointly affects cell differentiation and matrix production [97,98,99,100]. It is important to control the voltage and concentration during the electrospinning process to achieve the desired diameter and pore size, which plays a crucial role in the biocompatibility and long-term stability of the grafts in vivo.

#### 4.1.3. The Effects of Surface Roughness and Fiber Orientation on Cellular Behavior within Artificial Blood Vessels

Electrospun fibers form a matrix and an increase in the fiber diameter leads to an increase in the surface roughness of the matrix [93,100]. Surface morphology further influences cell behavior [101,102,103].

In a study, the effects of various fiber diameters (ranging from 0.11–3.4 μm) and surface average roughness values (ranging from 0.67–5.7 μm) on cell proliferation were investigated. The results demonstrate that larger PCL samples with higher Sa values exhibit enhanced cell proliferation rates [93]. However, conflicting results were also reported [104]. Given the limited research on the surface roughness of electrospinning, the optimal surface roughness setting requires further exploration by authoritative sources.

Furthermore, several studies observed that the alignment of nanofibers can also affect cellular behavior [105]. Ordered fiber alignment is more effective in promoting cell proliferation than randomly aligned fibers [106]. Cells migrate and align along the grooves on the substrate, appearing elongated on structures with an ordered fiber alignment [107]. Therefore, matrices containing fibers with an ordered alignment are more favorable for endothelial integration in artificial grafts [108].

In general, it was observed that higher surface roughness and the ordered alignment of nanofibers can have a beneficial effect on cellular behavior. On the textile fiber matrix, different diameters of spinning produce pores of different sizes, which further formed a matrix surface with a varying thickness and roughness. These three factors have a progressive relationship, providing the basis for cell adhesion and influencing cell behavior, thereby facilitating the achievement of the good biological performance of electrospun artificial grafts.

### 4.2. Modification of Electrospun Vascular Grafts’ Structure for Artificial Blood Vessels

#### 4.2.1. Modification of Electrospun Scaffold Structure for Reduced Thrombogenicity

Thrombosis is one of the main contributors for vascular graft failure. To reduce the risk of thrombosis, anticoagulant factors are incorporated into the graft through surface modifications and other methods to reduce platelet activation and adhesion in the blood vessels.

Heparin is a commonly used anticoagulant that inhibits the action of thrombin and coagulation factors in the blood [109]. A common method is to immobilize heparin on artificial vascular surfaces. Yao et al. used chitosan to immobilize heparin and achieved a reduction in platelet adhesion by increasing the amount of heparin immobilization through an increase in the proportion of chitosan in PCL/chitosan blends [110]. Some teams use PEG as a linker to immobilize heparin [111,112]. Another surface modification technique involves the use of inert gas plasma treatment. Li et al. utilized NH3 to plasma treat electrospun vascular structures [113]. There are also methods that use coating fixation and PEG amine linkers simultaneously [114].

To date, alternative anticoagulant methods have been developed to replace heparin. Chlorosulfonic acid can be used to treat SF to generate sulfonated SF, which greatly enhances the anticoagulant ability of the treated vessels. At the same time, it maintains a high expression of phenotypic markers and increases the proliferative capacity of EC and VSMC [115]. In addition, other approaches were proposed. Eldurini et al. developed a novel electrospinning scaffold using a wintergreen oil/PLC polymer solution [116]. Tran et al. grafted conjugated linoleic acid onto PU/PCL grafts via plasma treatment. The grafts were endowed with varying degrees of anticoagulation ability [117]. Overall, numerous modified methods aimed to reduce electrospun vascular thrombosis; however, further research and explorations are needed to assess the long-term effectiveness.

#### 4.2.2. Endothelialization Modification of Electrospun Structures

The endothelial layer has a significant role in the vascular system, promoting smooth vessel walls and anti-thrombotic properties. To date, the methods for improving endothelialization in electrospun vascular grafts can be roughly divided into three categories: using vascular endothelial growth factor (VEGF) and tyrosine kinase receptor to promote targeted endothelial cell activity, using stromal cell-derived factor 1α as a chemotactic agent to promote the aggregation of hematopoietic stem cells and utilizing endothelial progenitor cells, and another method is to improve it with small peptides composed of amino acids [118,119].

In a study utilizing a VEGF modification, VEGF and heparin were immobilized through a polydopamine (PDA) coating to improve endothelialization and enhance blood compatibility. The combined use of heparin and VEGF showed superior effects on behaviors, such as the proliferation and adhesion of HUVECs, compared to individual use [116]. Similarly, several teams developed electrospun vascular grafts with SDF1α coatings. Guo et al. studied VEGF/SDF1α PU electrospun grafts implanted in experimental dogs. They observed that SDF-1α and VEGF were rapidly released after implantation and had a significant effect on the promotion and differentiation of EPCs, thereby demonstrating the role of this growth factor [120]. Peptides composed of common amino acids are another type of improvement material that can help the adhesion of ECM proteins. ECM proteins contain an Arg-Gly-Asp (RGD) sequence, which facilitates recognition during cell adhesion [121]. Numerous methods exist for the improvement of endothelialization; however, some of them may have insufficient stability under physiological conditions or may be prone to toxicity, and some methods may be cumbersome to manufacture [122,123,124]. A simpler surface modification involves covalently immobilizing gelatin through photo-activated diazotized derivatives, resulting in a modified surface that exhibits excellent biological activity and blood compatibility [125].

## 5. Braid

Braid structures have limited applications in some traditional textile fields (such as the garment industry); however, they are widely used in industrial, sports, automotive, and other industries. Compared to other methods, the braid method started relatively late in the preparation of artificial grafts.

The braiding shown in Figure 5 is a fabric formed by using three or more yarns along the direction of the fabric formation, braided in a specific pattern (usually at a specific angle, etc.). In braiding, different parameters, such as braiding angle, braiding pattern, yarn quantity and diameter, and friction coefficient between yarns, have an important impact on the performance and long-term use of artificial blood vessels [126]. For example, the tilted-mesh structure generated by the braiding angle also enables the braided tube to obtain radial expansion. Many articles have explored and evaluated these parameters, with discussions on the effects of the braiding angle and yarn diameter and quantity being more extensive.

The unique structure of braided tubes allows for easy compressibility and recoil, as well as lower the bending stiffness. This technique not only facilitates the production of large-diameter grafts, but also provides an alternative approach for manufacturing small-diameter artificial grafts [127]. When transplanted in vivo, the movement of yarn at the intersection is restricted due to sutures at the joint of the braided artificial grafts, which may cause them to fail and break [128]. Moreover, the radial force of the braided scaffold is relatively insufficient and has a certain impact on the long-term support of blood vessels.

### 5.1. The Influence of Characteristics of Braided Structures on Artificial Blood Vessels

Several parameters of braiding artificial blood vessels can affect the graft’s performance, including vasodilation, inflammation levels, and cell survival rate, highlighting the importance of the research in this area [129]. According to the experimental models, blood vessel formation in the human body is accomplished through inflammation-driven and mechanically mediated processes [130]. Therefore, the researchers improved the chemical, mechanical, and morphological properties of scaffolds by altering the design parameters during the braiding process to regulate the degree of inflammation induction [131]. At the same time, the mechanical properties of grafts affect the stress shielding capacity of cells in neotissue, which, in turn, determines the production, remodeling, and degradation of ECM [132]. It can be seen that the study of factors, such as braiding angle, yarn count, and diameter, is of great importance; however, the research on the effects of some braiding patterns on the performance of artificial grafts is still limited.

#### 5.1.1. Influence of Braiding Angle on Mechanical Properties

The angle at which the yarn is braided can affect the overall mechanical performance of the artificial blood vessel, causing the vessel to exhibit different bending deformations and varying levels of fatigue resistance [133].

Numerous studies suggest that larger braiding angles are favorable for improving the mechanical performance of blood vessels. A team fabricated six groups of vessels with different braiding parameters (braiding angle, number, and diameter of metal wires) and used a finite-element analysis to study their mechanical properties. The results show that, under a pressure of 500 Pa with 24 wires braided, the minimum radial deformation decreases by 0.8175 mm as the braiding angle increases from 30° to 75°. In other words, the radial force of the artificial vessel increased and the deformation decreased with an increasing braiding angle [134]. Zheng et al. utilized computational models to investigate and discover that larger braiding angles of Nitinol and PET yarns can result in the better flexibility and radial strength of braided stents. Furthermore, the study suggested that the influence of braiding angles on various properties decreased when the braided stent experienced a significant deformation. Therefore, it is possible to enhance the radial mechanical performance of braided grafts by properly increasing the braiding angle. To better understand the subtle mechanical changes produced by smooth muscle cells in braided artificial grafts, the team utilized multi-scale finite-element codes. It was found that an artificial blood vessel composed of 12 PGA fibers with a braiding angle of 15 degrees could effectively promote the function of smooth muscle cells [135].

To improve the deformation caused by radial forces in braided blood vessels, a coating can be applied. Adding a polymer-based coating to a metal braided stent provides an improvement of the deformation. These coatings enabled the braid to obtain improved strength at all braiding angles [136].

#### 5.1.2. The Influence of the Number and Diameter of Braided Yarns on the Mechanical Properties and Porosity of a Material

In addition to the angle of yarn, increasing the number and diameter of braided yarns also enhanced the radial strength of the braided blood vessel. In an article by Zheng et al., they believed that the diameter of nickel–titanium wires had a greater impact on the mechanical performance of the stent compared to other parameters, and larger wire diameters resulted in stronger radial forces [136]. However, at the same time, an increase in the number and diameter of braided yarns decreased the porosity of the braided tube. Therefore, during the actual preparation, the number and wire diameter should be controlled at a relatively high level; however, not one that is excessively high. This balances the vascular strength and porosity, while avoiding a decrease in the vascular flexibility [134].

It is worth further investigating the relationship between the braiding angle mentioned in the previous section and the yarn diameter mentioned in this section, as they both have an impact on blood vessel performance. The bending moment of grafts is mainly affected by the braiding angle and yarn diameter, and a larger braiding angle can help achieve better flexibility, while an increase in yarn diameter may result in reduced flexibility but increased radial strength. Further detailed experimental data show that, as the bending angle increases from 30° (25%) to 60° (33%), the influence of the yarn diameter gradually surpasses that of the braiding angle [136]. Multiple influencing factors are interrelated and collectively affect the mechanical properties of blood vessels.

### 5.2. Modification of Braided Artificial Blood Vessels

This section mainly focused on the modifications for improving the biocompatibility of braided artificial blood vessels, which have various approaches. Recently, many teams have been working on degradable materials that show great potential in improving the biocompatibility of grafts. Silk protein, which has been mentioned before, is also widely used. In addition to some commonly used methods, other novel approaches were also studied.

In 2006, a team developed small-diameter degradable braided blood vessels with good biocompatibility using biomaterials. The vessel incorporated a layer of PCL coated onto the exterior of PGLA, forming a porous PCL–PGLA composite-material braided vessel. Experimental studies showed that fibroblasts proliferated in the PCL porous layer within a week, demonstrating good biocompatibility and facilitating tissue regeneration and degradation [137]. Moreover, the issue of the strength of degradable braided grafts is generally not a concern. Zhao Guoqiang et al. used PLLA fibers as raw materials to braid biodegradable artificial grafts. Experimental tests showed that the mechanical properties of the PLLA scaffold were good, and the radial force curve overlapped with that of a similar carotid artery wall stent [138]. To improve the mechanical properties of biodegradable polymer scaffolds, Deng et al. prepared a PLA braided stent coated with a six-arm PLCL coating, which effectively constrained the axial movement of individual filaments at cross points and enhanced the mechanical performance of the scaffold [139]. In recent years, SF has also become a commonly used material in artificial blood vessel engineering and can play a role in small-diameter blood vessel grafts [140,141]. Not only does it have good biocompatibility and controllable biodegradability, but it also exhibits excellent mechanical properties and minimal inflammatory responses [142]. A research team designed a novel SF biomimetic scaffold composed of three different layers, with the tubular braided layer in the middle and the inner and outer layers constructed through freeze-drying. Cell toxicity and compatibility were tested on L929 cells and a human umbilical vein, and the vascular scaffold exhibited good mechanical properties, permeability, and cell adhesion.

## 6. Conclusions

In summary, this paper reviewed the influence of structural characteristics on the vascular performance for four types of biomedical artificial blood vessels, and introduced various methods to improve their performance. In contrast to the other literature, this paper focused more systematically on the principles of textile structure, demonstrating how changes in textile details, such as warp knitting, weft knitting, fabric organization, thread diameter, and fiber angle, affect properties, including compliance, mechanical performance, porosity, and suture strength. This enables future artificial blood vessel designs to reduce current unreasonable designs by controlling the abovementioned parameters. Secondly, this article highlighted the advantages and disadvantages of different textile methods, including knitting, weaving, braiding, and electrospinning. This allows for the full utilization of each method’s strengths, as well as the possibility of composite multilayer designs for comprehensive use. In addition, this paper reviewed the methods used to optimize the vascular performance of different artificial grafts. Although some methods may be similar, a detailed distinction is beneficial for improving the rigor of different types of vascular modification. However, this paper did not provide a detailed description of the multilayer structure of textile grafts or the design of large-diameter grafts. Readers are advised to consult other sources for more information on these aspects.

Despite the extensive research on textile-based artificial grafts for medical use, there are still challenges and issues that need to be addressed. Firstly, the design of the fabric structure is the basis for blood vessel design; however, in the literature at present, a more thorough investigation is required to understand the specific relationship between fabric structures, braiding methods, and the multiple properties of grafts. The patterns used at present are relatively basic and do not fully utilize the unique advantages created by the intricate interlacing of textiles. Perhaps this could serve as a train of thought for optimizing the fundamentals of textile-based grafts. Furthermore, during the writing process of this article, it was also believed that the porosity of the fabric was not optimally utilized. In the 2D or 3D fabric design of textiles, it is important to maintain highly porous characteristics to facilitate nutrient and gas exchange, promote cell migration, and regenerate tissue, which is crucial for achieving the long-term biological stability of artificial grafts. Despite some relevant literature on maintaining porosity and preventing bleeding, further attention is needed to find good designs and solutions. Furthermore, continued attention and research are needed regarding the long-term patency of blood vessels. This issue involves the comprehensive consideration of multiple performance factors, such as the compliance, mechanics, biocompatibility, and antimicrobial properties of artificial blood vessels. In terms of biocompatibility, antibacterial properties, and infection resistance, various modification methods mentioned earlier, such as adding active factors, drug-releasing coatings, bioactive coatings, or using bio-based yarns, can promote blood vessel biological activity, reduce the risk of inflammation, vascular occlusion, etc. However, in terms of the mechanical performance, it is not enough to only consider the match between the blood vessel and human body under static conditions; dynamic adaptation in the living body should also be considered, while paying attention to compliance to simulate natural blood vessel behavior. In order to investigate and examine the natural performance of grafts, animal experiments were also conducted in the development of textile artificial grafts. However, due to differences, such as the endothelialization rate, the research on small animals still cannot fully simulate the human body. We hope that further attention will be given to transplantation experiments of textile artificial grafts in large animals, gradually achieving dynamic testing and helping to catalyze the development of artificial grafts.

To date, textile blood vessel engineering and tissue engineering techniques, such as active cell culture and seeding, are gradually merging. Meanwhile, the continuous improvement of traditional textile technology and the vigorous development of electrospinning technology are meeting the demand for small-diameter artificial blood vessels, making it a more promising research field. Despite the unresolved issues, the research on medical textile artificial blood vessels is flourishing, and vascular transplantation therapy for cardiovascular and other diseases is gradually advancing towards a more sophisticated direction. It is hoped that the researchers can pay more attention to the special effects created by textile structures and discover better modification methods that simulate natural blood vessels.

## Figures and Tables

**Figure 1 polymers-15-03003-f001:**
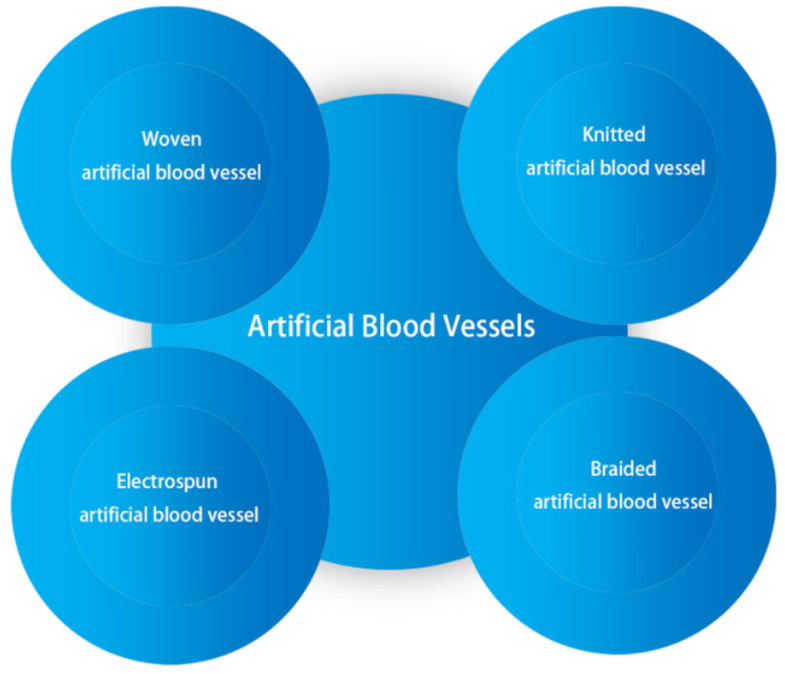
Classification of textile artificial blood vessels.

**Figure 2 polymers-15-03003-f002:**
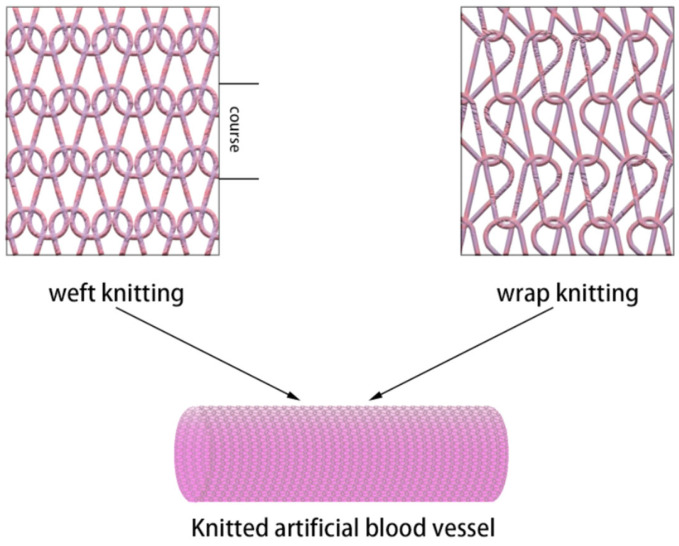
The structure of a knitted artificial blood vessel.

**Figure 3 polymers-15-03003-f003:**
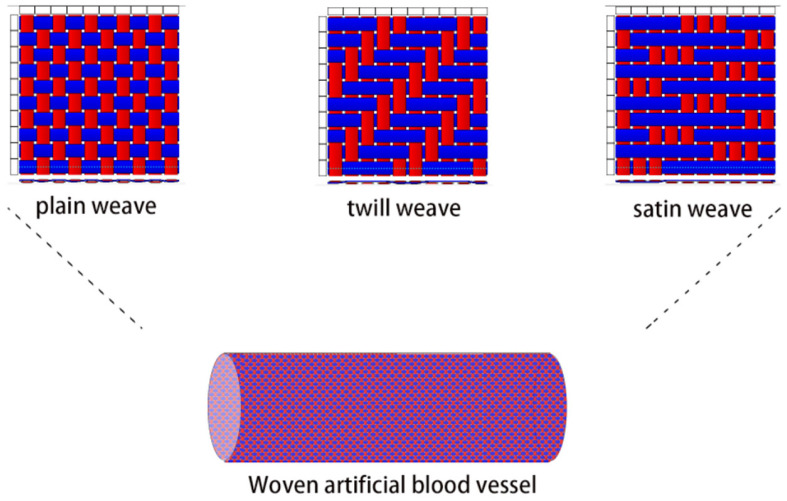
The structure of a woven artificial blood vessel.

**Figure 4 polymers-15-03003-f004:**
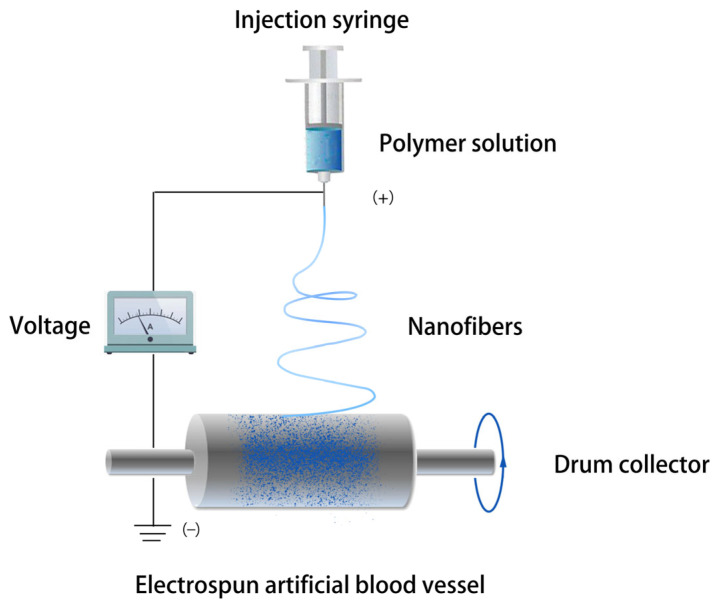
Preparation process of electrospun vascular grafts.

**Figure 5 polymers-15-03003-f005:**
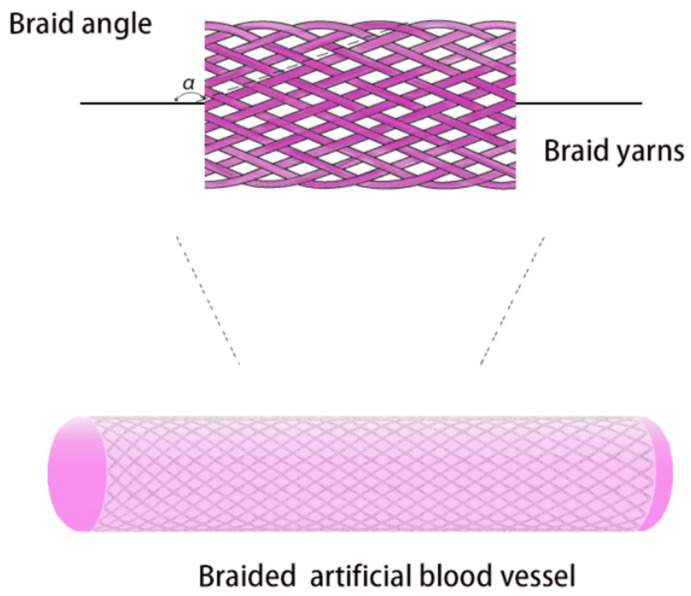
The structure of a braided artificial blood vessel.

**Table 1 polymers-15-03003-t001:** Information of main polymers used in preparation of artificial blood vessels by different textile methods.

Formation Technique	Polymers
Knitting	Polyvinyl alcohol (PVA)
	Polyglycolic acid (PGA)
	Polylactic acid (PLA)
	polyethylene glycol (PEG)
	Poly(l-lactide-co-ε-caprolactone) (PLCL)
	Polyethylene terephthalate (PET)
	Silk fibroin (SF)
	Collagen
Weaving	Polylactic acid (PLLA)
	Poly(ε-caprolactone) (PCL)
	Polytrimethylene terephthalate (PTT)
	Silk fibroin (SF)
Electrospinning	Polylactic acid (PLA)
	Polyglycolic acid (PGA)
	Poly(lactic-co-glycolic) acid (PLGA)
	Polycaprolactone (PCL)
	Poly(l-lactide-co-ε-caprolactone) (PLCL)
	Polyurethanes (PU)
	Silk fibroin (SF)
	Collagen
	Polysaccharides
Braiding	Poly(p-dioxanone) (PPDO)
	Poly(l-lactide-co-ε-caprolactone) (PLCL)
	Polycaprolactone (PCL)
	Polyglycolic acid (PGA)
	Polylactic acid (PLLA)
	Polyethylene terephthalate (PET)
	Poly(glycolide-co-lactide) (PGLA)
	Silk fibroin (SF)

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
