# Peer review of "The Influence of Textile Structure Characteristics on the Performance of Artificial Blood Vessels"

_polymers, 2023, doi:10.3390/polym15143003_

Round 1

Reviewer 1 Report

The subject of the paper is relevant and of high interest for the “Polimers” but the topic and the study may be more suitable for “Textiles”.

The following revisions need to be corrected.

1.            Abstract:

                    Consider adding a sentence summarizing the main findings or conclusions of the review

2.            Introduction:

                    Please provide more specific and up-to-date statistics or references regarding the incidence of cardiovascular disease and the annual treatment costs. This will significantly enhance the credibility and relevance of the information provided.

                    Discuss the specific challenges and limitations associated with the shortage of suitable donor blood vessels for cardiovascular disease treatment. Elaborate on the implications of this issue and its effects on patient outcomes.

                    Provide a concise overview of the advantages and disadvantages of alternative methods for manufacturing artificial blood vessels, such as 3D printing and hydrogel tube formation, in comparison to textile-based methods. This will highlight the unique benefits of textile-based approaches

                    Please include additional references to support the claims made regarding the advantages of textile artificial blood vessels in terms of flexibility, human characteristics, and permeability. This will significantly strengthen the argument and provide a more comprehensive understanding of the topic.

3.            Knitting:

                    Include additional information on the advantages and disadvantages of knitted artificial blood vessels. While you mentioned some advantages, such as extensibility, elasticity, and good conformity to the human body, it would be beneficial to discuss potential drawbacks or limitations as well.

                    When discussing modifications for improving biocompatibility, provide more specific information on the techniques used to enhance adhesion and endothelialization of knitted artificial blood vessels. Elaborate on the results and implications of the mentioned studies to highlight their significance.

                    Provide more specific details on the drug-loaded polymers or modified coatings used for enhancing antibacterial properties. Discuss the mechanisms of action and the specific bacteria targeted by these modifications.

                    Consider including information on the long-term performance and durability of knitted artificial blood vessels. Discuss any studies or data that demonstrate their stability and patency over extended periods.

4.            Weaving:

                    Organize paragraph structure, Some paragraphs in this section are quite long and detailed. Breaking them into smaller, focused segments can improve readability and enhance understanding of the information.

5.            Electrospinning:

·         Mention specific examples of synthetic and natural polymers commonly used in electrospinning for artificial blood vessels. Provide a brief description of their properties and how they contribute to the desired characteristics of the vessels.

·         Emphasize the unique advantages of electrospinning for artificial blood vessels, such as its ability to mimic the natural extracellular matrix structure, achieve multiple functionalities, and provide biocompatibility. Discuss how these advantages contribute to the overall performance and success of artificial blood vessels.

·         While discussing the advantages, also mention the limitations of electrospun non-woven artificial blood vessels, such as challenges in controlling pore size distribution and slightly inferior mechanical performance compared to woven vessels. This provides a balanced perspective on the technique.

·         Connect structure characteristics to performance: Further elaborate on the impact of electrospun structure characteristics (fiber diameter, porosity, fiber orientation, and surface morphology) on the performance of artificial blood vessels. Explain how each characteristic influences key factors such as endothelial cell activity, macrophage polarization, endothelial cell adhesion, and migration ability.

·         Include specific research findings or studies that have investigated the relationship between fiber diameter and endothelial cell activities.

6.            Braid:

·         Elaborate on the relationship between different braiding parameters and their impact on the mechanical properties and performance of artificial blood vessels. Explain how factors such as braiding angle, pattern, yarn quantity, and diameter influence vasodilation, inflammation levels, and cell survival rate. Provide specific examples and references to support these findings.

·         Provide a more detailed discussion on various approaches and techniques used to improve the biocompatibility of braided artificial blood vessels. Highlight the recent advancements in using degradable materials, such as silk protein and biodegradable polymers, and their positive impact on tissue regeneration and degradation. Include specific examples and experimental results to support these claims.

7.    Conclusion:

·         Emphasize the significance of optimizing fabric porosity to balance bleeding prevention during implantation and the benefits to vascular biology. Discuss the need for continued attention and research regarding the long-term patency of artificial blood vessels, considering multiple performance factors such as compliance, mechanics, biocompatibility, and antimicrobial properties.

Author Response

Dear editor,

Thank you very much for reviewing our manuscript. We have carefully revised our manuscript based on the reviewer's comments. Listed below are point-to-point responses to each of the comments. We are including a highlighted version of this manuscript indicating the changes that have been performed within this revised version. We hope that you will find our responses satisfactory and that the revised manuscript will be acceptable for publication in the Polymers. Again, we appreciate your time and efforts in the handling of our manuscript.

Sincerely,

Xingyou Hu, Ph.D.

Professor

Department of College of Textiles & Clothing, Qingdao University, Qing Dao, China.

No. ZR2021QC112

Title: The Influence of Textile Structure Characteristics on the Performance of Artificial Blood Vessels

Response to Reviewer 1

  1. Abstract:
  • Consider adding a sentence summarizing the main findings or conclusions of the review

Response: Based on your feedback, we have revised the abstract to make it more comprehensive.

  1. Introduction:
  • Please provide more specific and up-to-date statistics or references regarding the incidence of cardiovascular disease and the annual treatment costs. This will significantly enhance the credibility and relevance of the information provided.

Response: This is a great suggestion. We have continued to review the literature and added relevant data on incidence rates, medical costs, and other related information.

  • Discuss the specific challenges and limitations associated with the shortage of suitable donor blood vessels for cardiovascular disease treatment. Elaborate on the implications of this issue and its effects on patient outcomes.

Response: After receiving feedback, we have added information on the current challenges and prognostic impacts of autologous and allogeneic vessel transplantation in cardiovascular disease surgery.

  • Provide a concise overview of the advantages and disadvantages of alternative methods for manufacturing artificial blood vessels, such as 3D printing and hydrogel tube formation, in comparison to textile-based methods. This will highlight the unique benefits of textile-based approaches

Response: We appreciate the valuable suggestions from the reviewer. Additionally, we briefly highlight the advantages and disadvantages of other textile methods for producing artificial blood vessels to provide a comprehensive overview.

  • Please include additional references to support the claims made regarding the advantages of textile artificial blood vessels in terms of flexibility, human characteristics, and permeability. This will significantly strengthen the argument and provide a more comprehensive understanding of the topic.

Response: We continue to provide detailed descriptions and discussions regarding the various unique properties of textile vessels in the manuscript.

  1. Knitting:
  • Include additional information on the advantages and disadvantages of knitted artificial blood vessels. While you mentioned some advantages, such as extensibility, elasticity, and good conformity to the human body, it would be beneficial to discuss potential drawbacks or limitations as well.

Response: Following the discussion of the advantages of knitted blood vessels in the manuscript, we further explored two shortcomings.

  • When discussing modifications for improving biocompatibility, provide more specific information on the techniques used to enhance adhesion and endothelialization of knitted artificial blood vessels. Elaborate on the results and implications of the mentioned studies to highlight their significance.

Response: Regarding the discussion of modification, we expanded on the content, results, and impacts of relevant experiments and provided a thorough explanation.

  • Provide more specific details on the drug-loaded polymers or modified coatings used for enhancing antibacterial properties. Discuss the mechanisms of action and the specific bacteria targeted by these modifications.

Response: We included detailed information on the drug mechanism related to antibacterial modification.

  • Consider including information on the long-term performance and durability of knitted artificial blood vessels. Discuss any studies or data that demonstrate their stability and patency over extended periods.

Response: Thank you for this suggestion. We continued to review the literature and further discussed the favorable aspects of knitted blood vessels for long-term stability in the manuscript.

  1. Weaving:
  • Organize paragraph structure, Some paragraphs in this section are quite long and detailed. Breaking them into smaller, focused segments can improve readability and enhance understanding of the information.

Response: We have removed a portion of the woven content.

  1. Electrospinning:
  • Mention specific examples of synthetic and natural polymers commonly used in electrospinning for artificial blood vessels. Provide a brief description of their properties and how they contribute to the desired characteristics of the vessels.

Response: Regarding polymers, this manuscript has added a paragraph for discussion and enumeration.

  • Emphasize the unique advantages of electrospinning for artificial blood vessels, such as its ability to mimic the natural extracellular matrix structure, achieve multiple functionalities, and provide biocompatibility. Discuss how these advantages contribute to the overall performance and success of artificial blood vessels.

Response: We utilized relevant materials to further investigate the unique biological advantages of electrospun blood vessels and analyzed the reasons why they gain an advantage in terms of bioactivity.

  • While discussing the advantages, also mention the limitations of electrospun non-woven artificial blood vessels, such as challenges in controlling pore size distribution and slightly inferior mechanical performance compared to woven vessels. This provides a balanced perspective on the technique.

Response: Thank you for the reviewer's comments.

  • Connect structure characteristics to performance: Further elaborate on the impact of electrospun structure characteristics (fiber diameter, porosity, fiber orientation, and surface morphology) on the performance of artificial blood vessels. Explain how each characteristic influences key factors such as endothelial cell activity, macrophage polarization, endothelial cell adhesion, and migration ability.

Response: In each chapter, we further discussed and summarized the effects and interrelationships of fiber diameter, porosity, fiber orientation, and surface morphology on blood vessels.

  • Include specific research findings or studies that have investigated the relationship between fiber diameter and endothelial cell activities.

Response: Furthermore, we provided a more detailed explanation of the relationship between diameter and cell activity based on our previous descriptions.

  1. Braid:
  • Elaborate on the relationship between different braiding parameters and their impact on the mechanical properties and performance of artificial blood vessels. Explain how factors such as braiding angle, pattern, yarn quantity, and diameter influence vasodilation, inflammation levels, and cell survival rate. Provide specific examples and references to support these findings.

Response: In response to this suggestion, we have added new references to support the claim that multiple factors affect cell activity and vascular performance.

  • Provide a more detailed discussion on various approaches and techniques used to improve the biocompatibility of braided artificial blood vessels. Highlight the recent advancements in using degradable materials, such as silk protein and biodegradable polymers, and their positive impact on tissue regeneration and degradation. Include specific examples and experimental results to support these claims.

Response: Regarding the discussion of biocompatibility, we continued to summarize various studies and demonstrated their positive impacts.

  1. Conclusion:
  • Emphasize the significance of optimizing fabric porosity to balance bleeding prevention during implantation and the benefits to vascular biology. Discuss the need for continued attention and research regarding the long-term patency of artificial blood vessels, considering multiple performance factors such as compliance, mechanics, biocompatibility, and antimicrobial properties

Response: In the conclusion of the article, we added detailed discussion of porosity and the importance of multiple factors in vascular function.

Please refer to the attached document for the manuscript.

Sincerely,

Xingyou Hu, Ph.D.

Reviewer 2 Report

The present review reports four textile methods currently used in the manufacture of artificial blood vessels: knitting, weaving, electrospinning, and braiding. Although the manuscript covers an important topic, there are several issues that need to be addressed.

My remarks are detailed below:

The review does not provide in-depth discussion on the influence of textile structure characteristics on the performance of artificial blood vessels. The authors should provide a more detailed discussion on the comparison of these characteristics for one polymer, for example PCL or PLA. The authors should add a Table with information of used polymers in preparation of artificial blood vessels by different textile methods – this will be very useful.

Please, provide recommendations for future research as Future Directions – which of the mentioned methods are more prospective in such application.

Please correct the citing of references in the text – before full stop.

Once the abbreviation is entered, it should be used and not re-entered e.g., see lines 271, 272, 274, 275.

Author Response

Dear editor,

Thank you very much for reviewing our manuscript. We have carefully revised our manuscript based on the reviewer's comments. Listed below are point-to-point responses to each of the comments. We are including a highlighted version of this manuscript indicating the changes that have been performed within this revised version. We hope that you will find our responses satisfactory and that the revised manuscript will be acceptable for publication in the Polymers. Again, we appreciate your time and efforts in the handling of our manuscript.

Sincerely,

Xingyou Hu, Ph.D.

Professor

Department of College of Textiles & Clothing, Qingdao University, Qing Dao, China.

No. ZR2021QC112

Title: The Influence of Textile Structure Characteristics on the Per-formance of Artificial Blood Vessels

Response to Reviewer 2

1.The present review reports four textile methods currently used in the manufacture of artificial blood vessels: knitting, weaving, electrospinning, and braiding. Although the manuscript covers an important topic, there are several issues that need to be addressed.My remarks are detailed below:The review does not provide in-depth discussion on the influence of textile structure characteristics on the performance of artificial blood vessels. The authors should provide a more detailed discussion on the comparison of these characteristics for one polymer, for example PCL or PLA. The authors should add a Table with information of used polymers in preparation of artificial blood vessels by different textile methods -this will be very useful.

Response: We increased the depth of exploration into the impact of vascular structure on performance and added many discussions in each chapter. In addition, in the first chapter, a table of polymers used in different textile methods was added, along with discussions on certain polymers.

  1. Please, provide recommendations for future research as Future Directions - which of the mentioned methods are more prospective in such application

Response: The article added future research directions in the concluding section.

  1. Please correct the citing of references in the text-before full stop.

Response: The references have been listed before each punctuation mark.

4.Once the abbreviation is entered, it should be used and not re-entered e.g., see lines 271, 272,274,275.

Response: Thank you for the reviewer's reminders. We have eliminated redundancies and corrected any input errors.

Please refer to the attached document for the manuscript.

Sincerely,

Xingyou Hu, Ph.D.

Round 2

Reviewer 1 Report

I believe the authors have sufficiently revised the manuscript; thus, my opinion on the publication of the current version will be positive.

Thanks

Author Response

Dear editor,

Thank you very much for reviewing our manuscript. We have carefully revised our manuscript based on the reviewer's comments. Listed below are point-to-point responses to each of the comments. We are including a highlighted version of this manuscript indicating the changes that have been performed within this revised version. We hope that you will find our responses satisfactory and that the revised manuscript will be acceptable for publication in the Polymers. Again, we appreciate your time and efforts in the handling of our manuscript.

Sincerely,

Xingyou Hu, Ph.D.

Professor

Department of College of Textiles & Clothing, Qingdao University, Qing Dao, China.

No. ZR2021QC112

Title: The Influence of Textile Structure Characteristics on the Per-formance of Artificial Blood Vessels

Response to Reviewer 1: We greatly appreciate the reviewer's scrutiny and have made further revisions to the language of the manuscript.

Sincerely,

Xingyou Hu, Ph.D.